# *Pax6* mutant cerebral organoids partially recapitulate phenotypes of *Pax6* mutant mouse strains

**Nurfarhana Ferdaos**[1,2], **Sally Lowell**[3], **John O. Mason**[1,4]*

**1** Centre for Discovery Brain Sciences, University of Edinburgh, Edinburgh, United Kingdom, **2** Department of Pharmacology and Chemistry, Faculty of Pharmacy, UiTM Selangor, Shah Alam, Malaysia, **3** Centre for Regenerative Medicine, University of Edinburgh, Edinburgh, United Kingdom, **4** Simons Initiative for the Developing Brain, University of Edinburgh, Edinburgh, United Kingdom

* John.Mason@ed.ac.uk

**Data Availability Statement:** All relevant data are within the paper and its Supporting Information files.

**Funding:** NF was supported by a PhD studentship from the Ministry of Higher Education, Malaysia

## Abstract

Cerebral organoids show great promise as tools to unravel the complex mechanisms by which the mammalian brain develops during embryogenesis. We generated mouse cerebral organoids harbouring constitutive or conditional mutations in *Pax6*, which encodes a transcription factor with multiple important roles in brain development. By comparing the phenotypes of mutant organoids with the well-described phenotypes of *Pax6* mutant mouse embryos, we evaluated the extent to which cerebral organoids reproduce phenotypes previously described *in vivo*. Organoids lacking Pax6 showed multiple phenotypes associated with its activity in mice, including precocious neural differentiation, altered cell cycle and an increase in abventricular mitoses. Neural progenitors in both *Pax6* mutant and wild type control organoids cycled more slowly than their *in vivo* counterparts, but nonetheless we were able to identify clear changes to cell cycle attributable to the absence of Pax6. Our findings support the value of cerebral organoids as tools to explore mechanisms of brain development, complementing the use of mouse models.

## Introduction

Embryonic development of the mammalian brain has predominantly been studied in animal models, primarily mice. The recent advent of cerebral organoids—small, 3D organ rudiments grown from pluripotent stem in culture cells that closely resemble normal embryonic brain tissue—has provided a new way to study embryonic brain development *in vitro*, potentially allowing reduced dependence on animal models, in line with the principles of the 3Rs [1]. Although less widely used than human cerebral organoids, mouse cerebral organoids have considerable potential as research tools to unravel molecular mechanisms of brain development [2]. Furthermore, mouse organoids make it possible to directly compare *in vitro* phenotypes in with those found *in vivo*, something that is not possible in the human but which is essential to validate organoids as an experimental model.

(Grant G32486). The funders had no role in study design, data collection and analysis, decision to publish, or preparation of the manuscript.

**Competing interests:** The authors have declared that no competing interests exist.

**Abbreviations:** 4OHT, 4-hydroxy tamoxifen; EdU, 5-ethnyl-2'-deoxyuridine; CSF, cerebro-spinal fluid; ES, embryonic stem cells; GFP, green fluorescent protein; KSR, Knockout serum replacement; NE, neuroepithelium; cKO, conditional knockout; PH3, phospho-histone 3; SFEBq, serum free embryoid body quick; Tc, cell cycle length; Ts, S-phase length.

Pioneering studies reporting the growth of cerebral organoids from mouse ES cells showed that murine organoids contain neural progenitors and early neurons that express the expected marker genes and are arranged in a way that closely resembles embryonic mouse brain [3–5]. Well-characterised cellular behaviours, such as interkinetic nuclear migration were also observed [5]. However, it remains uncertain whether development in organoids involves the same molecular mechanisms and pathways that control mouse brain development *in vivo*. Much of our understanding of embryonic mouse brain developmental mechanisms is derived from the study of mutant strains harbouring either spontaneous or engineered mutations in genes whose products regulate developmental processes [6,7]. Many such mutant strains have been studied in considerable detail and the effects of such mutations are therefore well understood. If the molecular and cellular pathways that regulate embryonic mouse brain development also operate in cerebral organoids, we would expect to see similar phenotypes in organoids and mouse embryos harbouring identical mutations. To test this possibility, we chose the *Pax6* gene. *Pax6* encodes a transcription factor which acts as a high level regulator of cerebral cortical development. Its many known roles include control of progenitor proliferation, differentiation, cell specification, cortical patterning and neuronal migration [8,9]. Strains of mice carrying either constitutive *Pax6* null mutations or tissue-specific conditional *Pax6* mutations have been extensively studied, giving us a good understanding of many of Pax6's effects on mouse cerebral cortex development [10–12]. This provides an excellent point of comparison, allowing us to establish the extent to which Pax6's activities in mouse cerebral organoids resemble its well-characterised roles *in vivo*.

In the present study, we derived cerebral organoids from *Pax6*$^{-/-}$ mouse ES cells [13]. We also generated new mouse ES cell lines harbouring a conditional *Pax6* allele and used them to derive organoids containing mosaic cortical cell-specific *Pax6* mutations. This allowed us to measure the effects of Pax6 loss on cortical organoids by comparing Pax6- with neighbouring Pax6+ cells, analogous to the use of chimaeric embryos to characterise mutant phenotypes [14]. Examination of our *Pax6* mutant cerebral organoids revealed precocious differentiation of neural progenitors, an increase in the proportion of progenitors that divided away from the ventricular edge and altered rates of progenitor proliferation, all of which have been reported as phenotypes of *Pax6* mutant mice. This study therefore validates cerebral organoids as a suitable model system for identifying bone fide developmental phenotypes.

## Results

### *Pax6*$^{-/-}$ ES cells give rise to well-formed cerebral organoids

Cerebral organoids were grown following the protocol described in [4], in which aggregated mouse ES cells are treated with the Wnt antagonist IWP2 to promote telencephalic fate [3]. After 24 hours, Matrigel was added to promote epithelium formation and maintain tissue integrity (Fig 1A). To establish that the protocol worked in our hands, we first used a previously described ES cell line, Foxg1::Venus ES [4]. In this line, a cassette encoding the GFP variant Venus has been inserted into the *Foxg1* locus, such that it reports on the expression of *Foxg1*, a transcription factor that is normally expressed throughout the embryonic telencephalon from early stages. Venus expression therefore provides a convenient proxy to detect telencephalic cells within organoids. To determine the efficiency and reproducibility of cortical organoid differentiation with this protocol, we collected Foxg1::Venus organoids from three separate batches at day 8 of culture (approximately equivalent to E (embryonic day) 12.5 mouse embryos). DAPI staining revealed the presence of multiple neuroepithelial rosettes, mainly located near the outer edges of the organoids, (outlined by white lines, Fig 1B). Quantitation revealed that neuroepithelium constituted 41 ± 3.7% of total cerebral organoid tissue

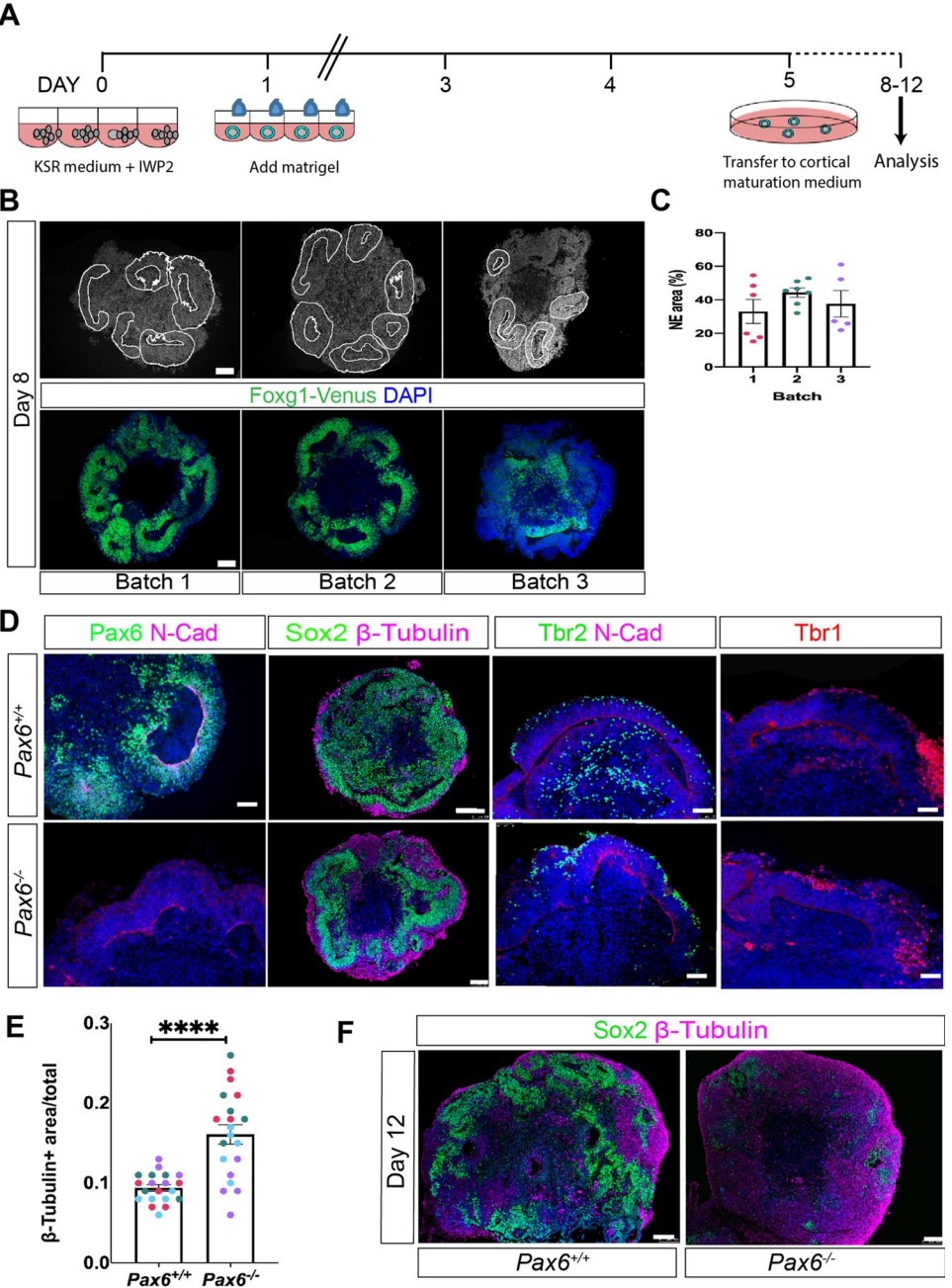

**Fig 1. Characterisation of *Pax6*-/- and *Pax6*+/+ (Foxg1::Venus) control cerebral organoids.** (A) Schematic outline of the organoid differentiation protocol (Eiraku et al., 2008). (B) Upper panels: DAPI-stained sections of day 8 organoids from three separate differentiation batches highlighting neuroepithelial regions of the organoids used for quantitation (outlined by white lines). Lower panels: Foxg1::Venus expression (green) in organoids from the same three batches. Nuclei are counterstained with DAPI (blue). (C) Graph showing proportion of organoid area that is occupied by neuroepithelial tissue. n = 5–6 organoids in each of 3 independent batches. Data points are colour-coded to indicate separate batches. (D) Immunostaining of day 8 cerebral organoids showing expression of neuroepithelial marker N-cadherin, basal progenitor and early neuronal marker Tbr2 (Eomes), cortical neuron marker Tbr1 and neuronal marker in *Pax6*+/+ control (top row) and *Pax6*-/- mutant organoids (bottom row). (E) Graph showing proportion of neuroepithelial area within day 8 *Pax6*-/- and control organoids that stained positively for β-tubulin+ expression. Coloured spots indicate separate batches of organoids. n = 20 control and 20 mutant organoids from 4 independent batches. Student's t-test indicated a significant increase in β-tubulin staining in mutant organoids, p<0.01. (F) Immunostaining for Sox2 and β-tubulin in day 12 *Pax6*-/- and control organoids. Scale bars in panels B,D,F: 50 μm.

(Fig 1C; n = 15 organoids from 3 independent batches). Immunostaining for GFP(Venus) indicated that the neuroepithelial area was Foxg1-positive, consistent with telencephalic identity (Fig 1B, lower panels).

To generate cerebral organoids lacking Pax6, we used the previously described ES cell line SeyD1 [13]. SeyD1 cells are homozygous for the *Pax6^SeyEd* null allele and are referred to as *Pax6^-/-* throughout. These cells contributed well to cortical tissue in chimaeric mouse embryos, and can readily be differentiated *in vitro* to form neural progenitors that show similar behaviour to *Pax6^-/-* primary progenitor cells in 2D culture [13]. *Pax6^-/-* SeyD1 ES cells successfully formed cerebral organoids which appeared broadly similar to controls (Fig 1D). We used immunostaining to characterise *Pax6^-/-* and control organoids on day 8 of culture for expression of key markers found in embryonic cortex (Fig 1D). As expected, Pax6 was expressed in neuroepithelium in control organoids (Fig 1D, upper panel). Pax6-expressing cells were mainly found close to the lumenal edge, equivalent to the ventricular edge in embryonic brains, where Pax6-expressing apical progenitors are located. Pax6 expression was absent in *Pax6^-/-* organoids (Fig 1D, lower panel). The neural stem cell marker Sox2 was widely expressed in neuroepithelial-like regions in organoids of both genotypes (Fig 1D). The transcription factor Tbr2/Eomes is expressed in basal progenitor cells and early postmitotic neurons in the embryonic cortex [14,15]. Tbr2-expressing cells were present in both control and *Pax6^-/-* organoids (Fig 1D), the majority of which were located toward the outer edge of the organoids, distant from the lumenal edge and similar to their location *in vivo*. Cells staining positively for Tbr1, a marker of early-born postmitotic cortical neurons were mainly located close to the outer edges of both control and *Pax6^-/-* organoids. In summary, the arrangement of cell types in organoids appeared similar to that found in the embryonic forebrain, with apical progenitor markers found close to the lumen, and basal progenitor and neuronal markers located more distally, toward the outer edge of the organoids.

In *Pax6^-/-* mutant mice, expression of the neuronal marker β-tubulin is significantly upregulated in E12.5 cortex compared to controls, indicating precocious neural differentiation [16]. We compared the expression of the neural progenitor marker Sox2 and differentiated neuron marker β-tubulin in *Pax6^-/-* mutant and control organoids on days 8 and 12 of culture. As described above, on day 8 both control and *Pax6^-/-* organoids contained extensive Sox2 + neuroepithelium and β-tubulin+ neurons were located basally, towards the outside of the organoids (Fig 1D). Quantitation confirmed a significantly greater area of β-tubulin+ staining in day 8 *Pax6^-/-* organoids (0.16 ± 0.01%), compared to controls (0.1 ± 0.004%) (Fig 1E, Student's t-test p = 0.0001, n = 20 organoids from 4 independent batches). *Pax6^-/-* organoids showed significantly greater variation in the extent of β-tubulin staining than controls did (F-test, p<0.001). By day 12, neuroepithelial structures were much less obvious, organoids of both genotypes appeared more disorganised and day 12 *Pax6^-/-* organoids showed greatly diminished expression of Sox2 and substantially more β-tubulin-positive tissue (Fig 1F), indicating that in the absence of Pax6, organoids show increased neural differentiation at early stages, as described in *Pax6^-/-* mutant mice [14,16].

## *Pax6^-/-* organoids exhibit changes in cell proliferation

Pax6 has multiple, well-described roles in regulating proliferation of cortical progenitors [11,14,16,17]. To look for evidence of cell cycle changes in *Pax6^-/-* organoids, we first determined the labelling index (LI) in day 9 organoids by pulse labelling with thymidine analogue 5-ethynyl-2'-deoxyuridine (EdU). Sections of EdU-labelled *Pax6^-/-* organoids and *Pax6^+/+* controls were stained for Tbr2 to identify regions of neuroepithelium with cortical identity. We found a significant decrease in the proportion of EdU+ cells in Tbr2-positive regions of *Pax6^-/-*

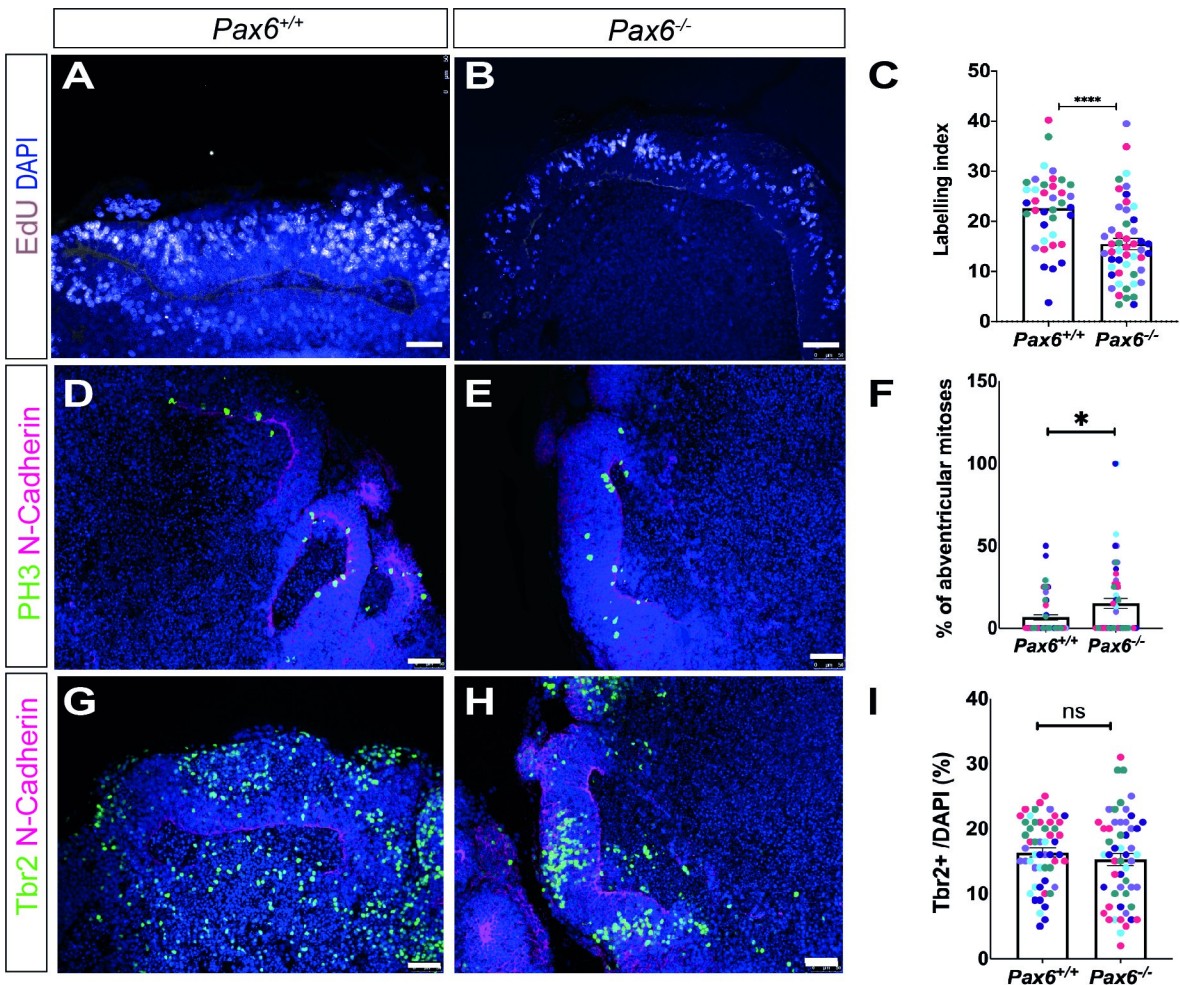

**Fig 2. Day 9 *Pax6*^-/- cerebral organoids display phenotypes similar to mouse *Pax6*^-/- cerebral cortex.** (A,B) *Pax6*^+/+ and *Pax6*^-/- organoids pulse labelled with EdU for 2 hours, stained for N-cadherin expression (red) and EdU (pink) (C) Graph showing EdU labelling index in control and *Pax6*^-/- day 9 organoids. LI is significantly lower in mutants (Student's t test, p<0.0001, n = 39–49 organoids from 5 batches). (D,E). Immunostaining for PH3 (green) to identify mitotic cells and N-cadherin (magenta) to show neuroepithelium. (F) Graph showing percentage of mitotic cells (PH3+) located 5 or more cell diameters away from lumenal edge in control and *Pax6*^-/- day 9 organoids. Mutants show a significant increase in abventricular mitoses (Student's t test, p = 0.0407, n = 28–43 organoids from 5 batches). (G,H) Immunostaining for Tbr2 (green) and N-cadherin (magenta) (I) Graph showing percentage of *Tbr2*+ cells in *Pax6*^-/- and control organoids. Student's t-test indicated no significant difference between mutants and controls, p = 0.381, n = 53–54 organoids from 5 batches). Graphs in panels C,F and I show mean ± SEM from 39–54 organoids of each genotype, consisting of ~10 organoids from each of 5 independent batches (indicated by spot colour). Scale bar: 50 μm.

organoids (15.1 ± 1.1%) compared to *Pax6*^+/+ controls (22.8 ±1.7%) (Fig 2A–2C, Student's t-test, p = 0.0001, n = 50 organoids of each genotype from 5 independent batches), indicating that loss of Pax6 altered progenitor proliferation.

Previous studies on *Pax6*^-/- mice have identified an increased number of mitotic cells, labelled by phospho-histone H3 (PH3), that are located away from the apical (ventricular) edge of the cortex, where mitosis of RGC progenitors normally occurs [14,16,17]. To determine whether loss of *Pax6* in cerebral organoids affects the location of mitotic cells, we stained organoids for PH3, counted the total number PH3+ cells in Tbr2-positive regions and the proportion that were located in an abventricular position, defined as five or more cell diameters away from the lumen edge. We found a clear increase in the number of abventricular mitoses

in $Pax6^{-/-}$ organoids (15.1 ± 2.6%) compared to control (6.3 ± 1.7%) (Fig 2D–2F, Students t-test, p = 0.041, n = 50 organoids from 5 independent batches). These proportions are similar to those previously described in E12.5 mouse embryos, ($Pax6^{+/+}$ 11%, $Pax6^{-/-}$ 16%) [16].

The developing cortex of $Pax6^{-/-}$ mice contains a decreased number of Tbr2+ cells [14]. We characterised the percentage of *Tbr2*+ cells, in day 9 $Pax6^{-/-}$ and control organoids and although there was a slight decrease in the proportion of *Tbr2*+ cells in mutants (14.6 ± 0.96%) compared to control (16.3 ± 0.7%) this change was not statistically significant (Fig 2G–2I, Student's t-test, p = 0.38, n = 50 organoids of each genotype from 5 batches).

## Conditional Pax6 mutant organoids exhibit cell cycle change

Conditional *Pax6* mutant mouse strains have been used to identify time- and tissue-specific effects of inactivating *Pax6* [11,12]. In one such previous study, we found that loss of Pax6 during early corticogenesis led to an increase in cortical progenitor proliferation and changes to the length of cell cycle (Tc) and S phase (Ts) [11]. The mouse strain used in the previous study contains a conditional (floxed) *Pax6* allele [18] together with an *Emx1-Cre^{ERT2}* transgene which drives cortex-specific expression of a tamoxifen-inducible form of cre recombinase [19] and the cre reporter allele, RCE [20]. This allows for timed, cortex-specific inactivation of *Pax6* and marks *Pax6*-deleted cells with a GFP fluorescent marker. No GFP+ cells were found in organoids that had not been treated with 4OHT, indicating an absence of 'leaky' cre activity.

To determine whether conditional Pax6 mutant cerebral organoids show phenotypes similar to those described in mice, we first derived two new ES cell lines with the same genotype as our *Pax6* conditional mutant mouse strain described above [11]. A male *Emx1-Cre^{ERT2}*;*Pax6^{fl/fl}*;*RCE* mouse was crossed with a *Pax6^{fl/+}* heterozygous female, blastocysts were collected on day 3.5 and treated with CHIR99021 and PD0325901 as described [21] to promote formation of ES cell lines. We established two ES cell lines with the genotype *Emx1-Cre^{ERT2}*;*Pax6^{fl/fl}*;*RCE* (i.e. homozygous for the conditional *Pax6* allele) and named them cKOhom1 and cKOhom2. Both lines had a normal karyotype and expressed the pluripotency markers Oct4 and Nanog.

Pax6 cKO cells gave rise to well-formed cerebral organoids, which contained Pax6-expressing neuroepithelium internally and β-tubulin+ neurons near the outer edge (Fig 3A and 3B). To produce cortical cell-specific deletion of Pax6, day 7 cKO organoids were treated with 1 μm 4-hydroxytamoxifen (4OHT) for 24 hours. This led to mosaic activation of the RCE cre reporter in the organoid neuroepithelia (Fig 3C and 3D), with 14 ± 3.1% of cells becoming GFP+ (Fig 3E). In situ hybridisation showed that the great majority of cells in the organoid neuroepithelia expressed *Emx1*, indicating that they have cortical identity (Fig 3F and 3G). This was true of both GFP+ cells, 92% of which were Emx1+ (Fig 3H) and GFP- cells. To determine whether the GFP+ cells had lost expression of Pax6, we co-stained cKO organoids for Pax6 and GFP (Fig 3I and 3J). Pax6 and GFP staining appeared to be mutually exclusive–98% of GFP+ cells did not detectably express Pax6 and a clear majority of GFP- cells were Pax6+ (Fig 3I and 3J). Taken together, these findings indicate that the neuroepithelial tissue in 4OHT treated organoids is mosaic–cells which express GFP have lost Pax6 protein, while GFP- cells retain Pax6. This allowed us to determine the effects of Pax6 loss on progenitor proliferation, by comparing cell cycle parameters between GFP+ (Pax6-) and GFP- (Pax6+) cells in the same areas of organoid cortical-like tissue, analogous to the use of $Pax6^{+/+}\leftrightarrow Pax6^{-/-}$ chimeric mice to identify cell-autonomous phenotypes in mutant cells (eg [14]).

4OHT treated cKO organoids were labelled with a two hour pulse of EdU to determine the labelling index (LI) of Pax6+ and Pax6- cells in cortical regions of organoids, identified by their GFP status. The LI in Pax6- cells (0.17 ± 0.02) was lower than that in Pax6+ cells (0.27 ± 0.02), (Fig 4A, n = 18 organoids from 3 batches, Student's t-test, p = 0.0006), similar to

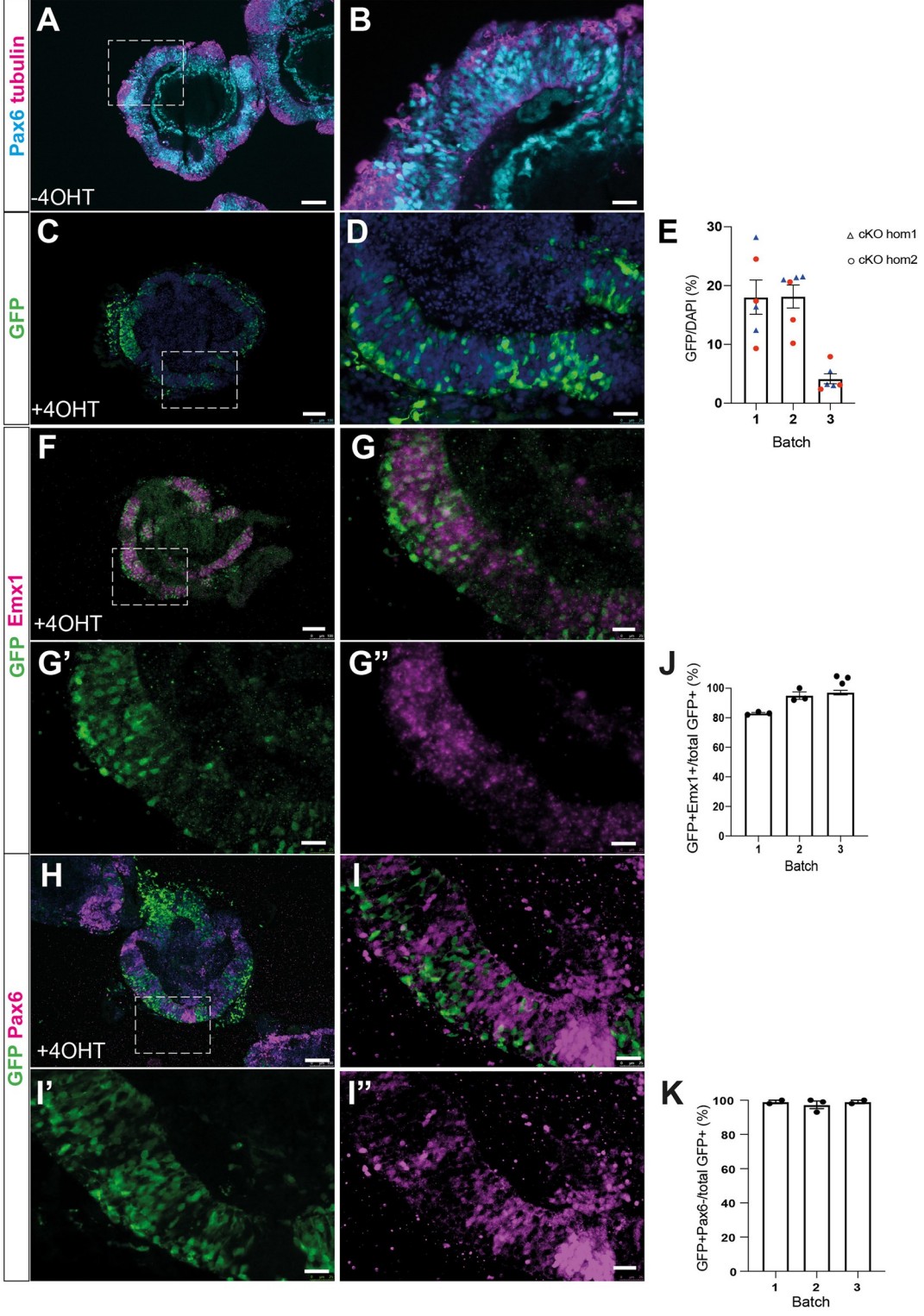

**Fig 3. Characterisation of *Pax6* cKO cerebral organoids.** (A,B) Pax6 and β-tubulin expression in day 8 cKOhom1 organoids treated with DMSO vehicle only. The boxed area in panel A is shown in B. (C,D) GFP staining of 4OHT treated organoids, showing mosaic activation of cre reporter in neuroepithelium. The boxed area in panel C is shown in D. (E) Quantitation showing efficiency of GFP-reporter activation in 4OHT treated cKOhom1 and cKOhom2 organoids. (F,G) Combined *in situ* hybridization for *Emx1* (magenta) and GFP immunostaining, showing mosaic activation of the cre-reporter RCE in cells with

cortical identity. Panels G' and G" show GFP and Emx1 channels separately. (J) Graph showing percentage of GFP+ cells that express cortical marker *Emx1* in three separate batches of 4OHT treated cKO organoids. (H,I) Co-staining for Pax6 and GFP indicates that the vast majority of GFP+ cells lack Pax6 expression in 4OHT treated CKO organoids. Panels I' and I" show GFP and Pax6 channels separately. (K) Graph showing percentage of GFP+ cells that do not express Pax6 in three separate batches of 4OHT treated cKO organoids. Scale bars A,C,F,H: 100 μm, B,D,G,G',G",I,I',I": 25 μm.

the finding in *Pax6$^{-/-}$* organoids (Fig 2C). We then determined the growth fraction (GF, the proportion of cells within the organoid neuroepithelial tissue that are actively proliferating) by staining for Ki67 and GFP (Fig 4B–4D) The GF of Pax6- cells was 0.71 ± 0.03 and for Pax6 + cells it was 0.64 ± 0.03. The difference was not significant (Student's t-test, p = 0.101, n = 4–6 organoids from each of two batches). Finally, we calculated lengths of the cell cycle (Tc) and S phase (Ts) in Pax6+ and Pax6- cells in *Pax6$^{cKO}$* organoids, using cumulative EdU labelling as described in [22] and employed in [16] to measure cell cycle parameters in *Pax6$^{-/-}$* mutant mice (Fig 4G and 4H). cKOhom1 and cKOhom2 day 8 cerebral organoids treated with 4OHT were given repeated doses of EdU at two hourly intervals over a ten hour period to cumulatively label proliferating cells. At each timepoint, organoids were collected, sectioned and stained for GFP and EdU. The proportions of GFP+(Pax6-) and GFP-(Pax6+) cells that were labelled with EdU were then calculated and plotted against the labelling period as described in [22]. The rate of uptake of EdU was slower in Pax6- cells than in Pax6+ cells (Fig 4G), indicating that Pax6- cells proliferated more slowly than their Pax6+ counterparts. This effect was found in organoids grown from each Pax6cKO line, as shown when data from cKOhom1 and cKOhom2 organoids was plotted separately (Fig 4H). Using this data to estimate the lengths of the cell cycle and S phase revealed that in Pax6+ cells in the organoids, Tc was 28.8 h and Ts was 8.99 h whereas in Pax6- cells, Tc was 48.3 h and Ts was 11.1 h (Fig 4I).

*In vivo*, cell cycle times of cortical progenitors vary significantly with position and developmental stage. In rostral cortex, where Pax6 expression levels are highest, we previously found that Tc was ~13.5 h and Ts ~6–7 h and in caudal cortex, where Pax6 levels are lower, Tc was ~10 h and Ts ~6–7 h [11]. Clearly, the values we found in organoids are considerably longer than those found *in vivo*, for both Pax6+ and Pax6- cells, indicating that the *in vitro* environment may have substantially slowed proliferation. Nonetheless, loss of Pax6 led to clear lengthening of the cell cycle in organoids. This is in contrast to the shorter cell cycle seen in *Pax6$^{-/-}$* mutants *in vivo*. However, the effects of Pax6 loss on cell cycle *in vivo* are complex, and show both developmental stage- and tissue-dependent variation—this is discussed more fully below. Taken together, our findings clearly indicate that multiple aspects of *Pax6$^{-/-}$* mutant phenotypes previously described in mice are also found in *Pax6$^{-/-}$* cerebral organoids.

## Discussion

Here, we compared phenotypes caused by the loss of Pax6 function in cerebral organoids with well-characterised phenotypes in cognate mutant mice and found a number of strong similarities. This supports the notion that the molecular mechanisms that control embryonic development of the forebrain *in vivo* also operate in organoids, strengthening the case for the use of organoids as tools to investigate these mechanisms.

*Pax6$^{-/-}$* organoids demonstrated precocious neural development, as previously found in *Pax6$^{-/-}$* mice [14,16]. This phenotype is thought to arise as a consequence of an increase in the number of neural progenitors that exit the cell cycle in *Pax6$^{-/-}$* mutants [14]. Interestingly, we saw a much larger variation in the amount of β-tubulin+ neurons produced in *Pax6$^{-/-}$* organoids than in controls. One possible explanation for this is that Pax6 acts to protect cells from the influence of signalling molecules in the extracellular environment as has recently been

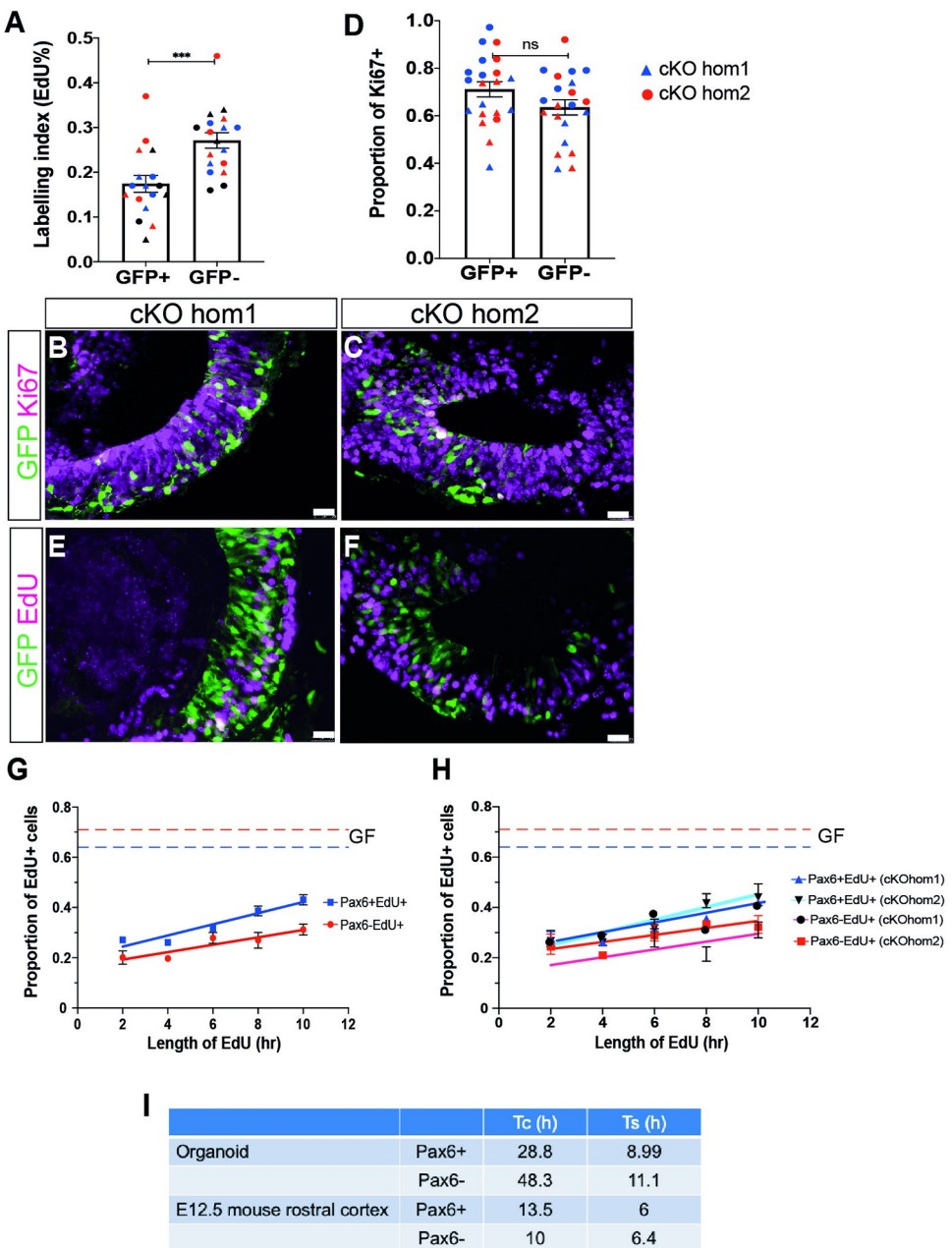

**Fig 4. Analysis of cell cycle parameters in *Pax6^cKO* cerebral organoids.** (A) Labelling index in GFP+ (Pax6-) and GFP- (Pax6+) cells in 4OHT treated cKO organoid neuroepithelium following a two hour pulse of EdU label. Each data point represents 1 organoid. Both cKO lines combined, n = 9 organoids for each line, consisting of 3 organoids from each of 3 independent batches (colour coded red, blue & black). (B,C) Ki67 staining of 4OHT treated cKOhom1 and cKOhom2 organoids to identify actively proliferating cells. (D) Graph showing proportions of Ki67 positive GFP + and GFP- cells in 4OHT-treated Pax6cKO organoids on day 8. n = 21 organoids in total, 4–6 organoids from each of two batches of organoids grown from each line (batches colour coded red and blue, triangles indicate cKOhom1 and circles cKOhom2). (E,F) Representative images of 4OHT-treated cKO organoids labelled with a 2 hour pulse of EdU. (G,H) EdU cumulative labelling show an increase in the proportions of EdU+ cells in both GFP+ and GFP- cells. Data presented as means ± SEM (n = 6 batches, 3 batches per line). In (G), cKOhom1 and cKOhom2 data are combined, these are shown separately in (H). (I) Table summarising Tc and Ts values in Pax6+ and Pax6- cells in organoids and E12.5 mouse cortex.

shown in mice [23]. It is possible that the variation in the extent of precocious neural differentiation in *Pax6*[-/-] organoids is due to variability in the presence of environmental signals that promote cell cycle exit. In controls, Pax6 would act to suppress cells' responses to such signals.

We found an increased number of abventricular mitoses in *Pax6*[-/-] organoids, another phenotype that has been previously described in E12.5 *Pax6*[-/-] embryonic cortex [14,16]. These abventricular mitoses could be due to disruption of interkinetic nuclear migration (IKNM), the process by which RGC nuclei move to the ventricular surface after leaving S-phase before undergoing mitosis. Alternatively, they could be due to an increased number of basal progenitor cells.

We found no change in the number of Tbr2+ cells in *Pax6*[-/-] organoids. At first sight, this appears to differ from previous work reporting an overall decrease in the number of Tbr2 + cells in *Pax6*[-/-] cortex at E12.5 [14]. However, the magnitude of the decrease reported previously [14] varied regionally across the cortex. Lateral cortex showed a large decrease in Tbr2 + cells; central cortex a modest, but significant decrease, and dorsal cortex no significant decrease. Therefore, if our organoids most closely resemble dorsal cortical tissue, our findings are consistent with the mouse phenotype. The question of the regional identity of cortical cells in our organoids is discussed below.

We found that the EdU labelling index (LI), was decreased in *Pax6*[-/-] organoids compared to controls, indicating that mutant cells proliferated more slowly. To identify consequences of acute, cortical cell-specific loss of Pax6, we generated two new ES cell lines homozygous for a conditional allele of *Pax6* and carrying an *Emx1-Cre*[ERT2] transgene and a GFP reporter cassette from a previously described *Pax6* conditional mouse strain [11]. We found that 4OHT treatment of the *Pax6*[cko] organoids grown from these cells led to mosaic loss of Pax6 in *Emx1*-positive regions. It is possible that the mosaic inactivation was due to poor penetration of 4OHT, perhaps due to the Matrigel coating of the organoids. Treatment with higher concentrations of 4OHT or treatment for longer periods were toxic to the organoids (not shown). The mosaic nature of this mutation allowed us to measure the effects of Pax6 loss on cortical cells in the organoids by comparing with Pax6+ cells in neighbouring cells, analogous to the use of chimaeric mouse embryos to characterise mutant phenotypes [14]. Comparing the behaviour of Pax6+ and Pax6- cells in the same organoid has the significant advantage of removing potential confounding effects of variation in cell behaviour between different clonal ES cell lines or organoid batches. Loss of Pax6 had no effect on the proportion of actively proliferating cells in the organoids which was similar to the proportion of proliferating cells reported in human cerebral organoids [24].

Cumulative EdU labelling of *Pax6*[cKO] organoids revealed significant lengthening both of the cell cycle time, from 28.8 hours in Pax6+ cells to 48.3 hours in Pax6- and of S-phase, from 9 hours to 11.9 hours in Pax6- cells. These times are substantially longer than found in E12.5 cortex *in vivo*, for both Pax6+ and Pax6- cells (w.t. Tc = 13.5 hr, Ts = 6 hr; *Pax6*[-/-] Tc = 10 hr, Ts = 6.4 hr [11,16]. The finding that cell cycle times are longer in both control and *Pax6*[cko] cells, together with the reduced EdU labelling index seen in *Pax6*[-/-] organoids suggests that the culture conditions may be affecting proliferation of all cells in the organoids. For example, in the absence of blood vessels and a normal circulation as found *in vivo*, access of nutrients, oxygen and growth factors to the interior of the organoids may be insufficient to support normal proliferation rates and the absence of proliferation-promoting factors normally found in CSF could be a factor [25]. There is evidence from studies of human cerebral organoids that culture conditions can affect proliferation [26,27] and single cell RNA seq analysis of human cerebral organoids indicated that many cells showed signs of metabolic stress, and have activated glycolysis, perhaps due to sub-optimal oxygen levels [28]. In human cerebral organoids, this issue has been addressed in a number of different ways, including increasing agitation in the

cultures through the use of bioreactors or culturing organoids on shaking platforms [29,30], cutting organoids open to allow growth medium to penetrate to the centre [31], or culturing thick slices of organoids at an air/liquid interface [32]. It would be interesting to test these approaches on mouse organoids too.

It is well-established that Pax6 regulates proliferation of cortical progenitors [11,14,16,17,33–37], so we expected to see differences in cell cycle times in organoid cells lacking Pax6. Pax6's effects on the cell cycle *in vivo* are highly context-dependent. In particular, levels of Pax6 expression influence its effects on progenitor cell cycle parameters. For example, in rostral E12.5 cortex, where Pax6 is expressed at high levels, *Pax6*$^{-/-}$ mutant embryos show a clear decrease in cell cycle length, whereas in caudal cortex, which expresses lower levels of Pax6, there is no difference in cell cycle length between *Pax6*$^{-/-}$ mutants and controls [11]. Similarly, E13.5 cortex-specific conditional *Pax6* mutants show a clear rostro-caudal gradient in the size of the effect of Pax6 loss on cell cycle [11]. We found increased cell cycle times in Pax6- cells in the organoids, in contrast to the effect of Pax6 loss in rostral cortex *in vivo*. As organoids lack the rostrocaudal and mediolateral axes required for normal cortical development, it is unclear to what extent regional phenotypes found in mice can meaningfully be mapped onto organoids. Further, as the gradient of Pax6 expression found in embryonic cortex [34,38] is important for establishing regional identity across the cortex [38,39] it remains possible that there are differences in regional cell identity in Pax6 mutant organoids, potentially accounting for the divergent effect of Pax6 on cell cycle. In future studies, single cell RNA seq will provide more information about the regional identities of cells in our organoids, and may highlight any differences in gene expression that could account for the different effect of Pax6 loss on progenitor proliferation. It is also worth noting that the ES lines used in this study differed in their genetic backgrounds, as detailed in Experimental procedures. It is possible that variations in genetic background may have contributed to some of the phenotypes we have reported here. However, any such contribution is likely to be very minor as as *Pax6*$^{-/-}$ mutant mouse brain phenotypes are very consistent across multiple genetic backgrounds including CD1 [16] and C57BL/6J-DBA/2J [40,41] and even the *Pax6*$^{-/-}$ rat phenotype is remarkably similar [42]. In conclusion, it is clear that mouse organoids can demonstrate clear phenotypes caused by mutations in developmental regulatory genes and are likely to be of value in studies of forebrain development, complementing *in vivo* studies.

## Experimental procedures

### Ethics statement

All experimental procedures involving mice were regulated by the University of Edinburgh Animal Welfare and Ethical Review Body in accordance with the UK Animals (Scientific Procedures) Act 1986 (licensing number P53864D41). Mice were sacrificed by cervical dislocation, no anaesthesia or analgesia were used.

### Mouse ES cell lines and culture

Foxg1::Venus ES cells [4] were purchased from RIKEN Bioresource Centre, Tsukuba, Japan (reference RBRC-AES0173). These cells are from an EB3 mouse genetic background. SeyD1 *Pax6*$^{-/-}$ ES cells (129SC/Ola background) were described previously [13]. To obtain *Pax6*$^{cKO}$ ES cell lines (*Emx1-Cre*$^{ERT2}$*; Pax6*$^{fl/fl}$*; RCE*), a male mouse carrying an *Emx1-Cre*$^{ERT2}$ allele [19], the GFP reporter allele RCE [20] and homozygous for the *Pax6*$^{fl}$ floxed allele [18] was crossed with a *Pax6*$^{fl}$ female heterozygote. Both parental mice were from a mixed CD1/C57Bl6 genetic background. Resultant blastocysts were collected at E3.5 and treated with CHIR99021 and PD0325901 (2i) as described in [21] to promote outgrowth of ES cell lines. Karyotyping

confirmed that newly derived ES lines were euploid and immunostaining for Oct4 and Nanog indicated pluripotency. All cell lines were routinely screened for mycoplasma contamination. All experimental procedures involving mice were regulated by the University of Edinburgh Animal Welfare and Ethical Review Body in accordance with the UK Animals (Scientific Procedures) Act 1986 (licensing number P53864D41). Mice were sacrificed by cervical dislocation, no anaesthesia or analgesia were used.

ES cells were cultured at 37°C and 5% $CO_2$ in Glasgow minimal essential medium (GMEM) supplemented with 1 mM sodium pyruvate, 2 mM glutamine, 1x non-essential amino acids, 0.1 mM β-mercaptoethanol, 10% fetal calf serum (FCS) and 100 units/ml leukemia inhibitory factor (LIF) on T25 flasks coated with 1% porcine gelatin. *Pax6$^{cKO}$* ES cell lines were maintained in the same medium, supplemented with 0.5 μM PD 0325901 (Axon Medchem).

## Cerebral organoid culture

Cerebral organoids were grown using the SFEBq (serum-free culture of embroid body-like aggregates–quick) protocol described in [4]. Briefly, ES cells were dissociated into single cells using Tryple (Invitrogen) and resuspended in KSR medium supplemented with 2.5 μM IWP2 (Sigma) where KSR (knockout serum replacement) medium consists of GMEM medium supplemented with 10% KSR, 1 mM sodium pyruvate, 0.1 mM non-essential amino acids and 0.1 mM β–mercaptoethanol. Aliquots of 5000 cells were plated into individual wells of low-adhesion U-shaped 96 well plates (Sumitomo Bakelite) and incubated at 37°C in 5% $CO_2$. Cells spontaneously formed aggregates overnight, which were then embedded in 200 μg/ml Matrigel with reduced growth factors (Corning) and transferred to 50 mm bacterial-grade Petri dishes (Fisher scientific). The Matrigel-coated aggregates were then cultured in cortical maturation medium (CMM) containing DMEM/F-12 medium supplemented with 1x N2 and 1x Glutamax (Thermo Fisher). Medium was replenished every other day.

## Labelling index and cumulative EdU labelling assay

Proliferating cells were labelled by adding 10 mM EdU (5-ethynyl-2'-deoxyuridine, Thermo Fisher) to the organoid culture medium. To measure the labelling index, organoids were incubated in the presence of EdU for 2 hours then fixed and stained. For cumulative EdU labeling assays, organoids were cultured in EdU-containing medium for 2, 4, 6, 8 and 10 hours. Labelled cells were subsequently detected by immunohistochemistry using the Click-iT EdU Alexa Fluor 647 Imaging Kit (Thermo Fisher Scientific) following manufacturer's instructions. Cell cycle length (Tc) and S-phase length (Ts) were calculated as described in [16,22].

## Immunofluorescence and fluorescent in situ hybridization

Organoids were washed twice with PBS and fixed with 4% paraformaldehyde (PFA) in PBS for 20 minutes at room temperature with shaking then cryoprotected in 30% sucrose solution at 4°C overnight. Sucrose was replaced with embedding medium (50:50 mixture of 30% sucrose: OCT medium) at 4°C for one hour and transferred to an embedding mould. Organoids were cryosectioned at 10 μm in a frozen microtome chamber (SakuraTek) and cryosections were collected on superfrost slides. Cryosections were stained with primary antibodies against N-Cadherin (BD Bioscience, 610920), Sox2 (Abcam, ab92494), Pax6 (Biolegend, 901301), Tbr2 (Abcam, ab183991), Tbr1 (Abcam, ab31940), GFP (Abcam, ab13970), PH3 (Abcam) or Ki67 (Abcam, ab15580). Subsequently, cryosections were incubated with appropriate secondary antibodies (Molecular Probes), stained with DAPI and mounted with Vectashield hard set (Vectorlabs). Negative controls omitting primary antibodies were included in all experiments.

Fluorescent *in situ* hybridsation was carried out as described in [43] using a probe specific for *Emx1* [44].

## Quantitation

Quantitative analysis of organoids was performed on histological sections using ImageJ. Quantification was restricted to neuroepithelial (NE) structures, defined by N-cadherin staining surrounding lumenal spaces. NE structures were delineated using the freehand selection tool as shown in Fig 1B (lower panels). For quantitative immunofluorescence analysis by area, images were separated by channel and the total area of organoid sections was determined based on DAPI staining (blue channel). Images were thresholded for each relevant channel and positively-stained areas were selected manually then measured in ImageJ. Quantification of cell numbers stained with the primary antibodies was restricted to neuroepithelial tissue within a 40x image, using the cell counter plug-in in ImageJ. To quantitate abventricular mitoses, total NE area was measured, the total number of PH3+ cells counted and the density of PH3+ cells per unit area calculated. The density of abventricular PH3+ cells (defined as >3 cell diameters from ventricular edge) was then calculated in the same way and the ratio of abventricular to total PH3+ cell density used as a measure of the proportion of abventricular mitoses.

## Supporting information

**S1 Table. Underlying data for Fig 1.**
(XLSX)

**S2 Table. Underlying data for Fig 2.**
(XLSX)

**S3 Table. Underlying data for Fig 3.**
(XLSX)

**S4 Table. Underlying data for Fig 4.**
(XLSX)

## Acknowledgments

We are grateful to Chiara Asselborn and Craig Murray for their contributions to the project, to Michael Molinek for invaluable assistance with deriving *Pax6^{cKO}* ES cell lines and to David Price for helpful comments on the manuscript.

## Author Contributions

**Conceptualization:** John O. Mason.

**Investigation:** Nurfarhana Ferdaos.

**Methodology:** Nurfarhana Ferdaos.

**Project administration:** John O. Mason.

**Supervision:** Sally Lowell, John O. Mason.

**Writing – original draft:** Nurfarhana Ferdaos.

**Writing – review & editing:** Sally Lowell, John O. Mason.

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
