## [Decision Letter · Decision Letter 0]

27 Sep 2022

PONE-D-22-20717Pax6 mutant cerebral organoids partially recapitulate phenotypes of Pax6 mutant mouse strainsPLOS ONE

Dear Dr. Mason,

Thank you for submitting your manuscript to PLOS ONE. After careful consideration, we feel that it has merit but does not fully meet PLOS ONE’s publication criteria as it currently stands. Therefore, we invite you to submit a revised version of the manuscript that addresses the points raised during the review process.

We look forward to receiving your revised manuscript.

Kind regards,

Michael Klymkowsky, Ph.D.

Academic Editor

PLOS ONE

Journal Requirements:

Additional Editor Comments:

There are a few minor comments that should be addressed (from one of the reviewers).  Please modify the text and let me know your response to them, and I think that this can manuscript be accepted without further review.  

Reviewers' comments:

Reviewer's Responses to Questions

**Comments to the Author**

1. Is the manuscript technically sound, and do the data support the conclusions?

Reviewer #1: Partly

Reviewer #2: Yes

2. Has the statistical analysis been performed appropriately and rigorously? 

Reviewer #1: Yes

Reviewer #2: Yes

3. Have the authors made all data underlying the findings in their manuscript fully available?

Reviewer #1: Yes

Reviewer #2: No

4. Is the manuscript presented in an intelligible fashion and written in standard English?

Reviewer #1: Yes

Reviewer #2: Yes

5. Review Comments to the Author

Reviewer #1: This manuscript describes molecular and cellular analyses of mouse forebrain organoids, comparing wild-type with those derived from Pax6-null ES cells, and comparing wt and Pax6-null cells in conditional mosaics. In general the results are similar to those seen previously in vivo - that the patterning of the telencephalon is broadly OK in Pax6 mutants, but with evidence of a slower cell cycle in Pax6-null cells and premature exit from the cell cycle, disturbing the proliferation-differentiation balance. The authors argue that the similarity between mutation effect in vivo and in the organoids validates the use of organoids for developmental studies. The experiments have been performed and presented well and the differences between organoid batches treated and described carefully. I LOVE the segregation of mutant and wild-type cells into stripes in the mosaic organoids, which reflects the in vivo chimeras. I have some technical and other comments below.

Technical things to be addressed:

1) Nothing is said of the genetic background of the ES cells used to make the Pax6<sey sey=""> organoids and whether they are the same as the control ES cells. If they are different, this could potentially undermine aspects of the quantitative analyses between the wt and null cell lines (in the same way that comparison between different mouse strains would be questionable). Authors should clarify and also describe genetic background of the conditional mutant cells.

2) In Figure 2 and lines 171-172, the reduction of EdU-positive cells with Pax6 mutation has a fairly marginal level of significance (P=0.043), which the authors need to justify or comment on. Also in Fig 2C this difference is represented as **** which seems excessive - * at best. This could be in inter-strain difference if genetic backgrounds are different (see 1). In fig 2F/line 196 the P value of 0.023 is better, but could still be in the range of inter-strain differences raised in 1).

3) Authors are to be commended for showing all data points in full, but the data in Fig 2F look weird to me. While there is a difference in mean % between wt and null, the data seem to consist of a lot oof organoids where no abventricular mitoses occurred, and a lot where roughly similar (comparing wt and null) levels of mitoses happened, and one outlier at 100%. The significant difference could be due to different proportion of organoids scored (wt v Pax6) that had no abventricular mitoses. These data need more critical comment

4) fig 3/4. It is difficult for the reader to compare colocalisation of GFP with nuclear markers e.g. EdU or with the Emx1 in situ. Can the authors present separate channels, and/or comment on how a magenta label was assigned to a patch of green GFP. The magenta channels in Fig 4B-F in particular are very weak, and I would have difficulty interpreting.

5) Can authors present or describe (line 226) whether they observed leakiness of Cre activity in absence of tamoxifen.

6) No negative controls are shown or described for any immunos or the in situ - should be addressed.

Self-important reviewer comments that authors might want to comment on or address:

7) The argument that it is unclear whether organoid development reflects the developmental pathways observed in vivo (lines 62-63) seems weak to me. It is very well established that multiple organoid systems from mice and humans self-organise and develop in very similar ways using very similar developmental pathways to in vivo. Sure there are differences due to absence of vascularisation and reduced complexity of cell composition, but the validity of organoids to inform in vivo development is not really in question. This does not detract from the value of the current study.

8) The authors comment on difference in cell cycle time and S-phase between organoids and in vivo. The discussion of regionalisation is important and valuable. Can the authors justify why they compare their data to E12.5 mouse and not later stages? Also I can understand why cell cycle time changes because cells might spend more time in G0/G1, but it is not clear to me why S-phase would increase - what is the mechanism for slowing down DNA replication? Is it firing of origins? It may not be relevant to current study, but authors could comment if they choose. Also the authors reference how Tc was calculated but the paper would benefit from description in methods.

9) Line 336 the authors comment that it was unexpected for the 4-OH tamoxifen to produce a mosaic deletion, but I would be astonished if it didn't. As far as I can tell, Tmx always produces mosaicism and the people who claim 100% recombination are fooling themselves. The mosaicism is a strength of the current study and the authors would be justified in saying they planned for this all along.

10) The morphology of the organoids is OK (adequate for the data), but I've seen a lot better. Do the authors want to comment on whether they consider their conditions optimised?

Minor comments:

11) Red and magenta are poor choices for presenting two-colour data (Fig 2A,B).

12) There is a problem with labelling of Y-axis in Fig 4H</sey>

Reviewer #2: The authors state that data is fully available without restriction but they don't describe where or how the data can be accessed, as required in the data availability statement.

This manuscript describes some carefully conducted experiments with mouse organoids designed to demonstrate the suitability of these models for studying gene expression changes by comparing the effects with those seen in animals with such manipulations. The gene chosen is the well characterised transcription factor PAX6, and as this study comes from a Lab that has made many contributions to our understanding of the role of PAX6 in cortical development. They have been able to adapt experimental measures used in animal studies such as neuron production, and progenitor proliferation including rates of proliferation. I am satisfied that there conclusion that the behaviour of PAX6 expressing cells in both embryonic cortex in vivo and in organoid preparations share strong enough similarities to make use of organoids a useful model in this context. Some differences and unexpected results were also found, and these are explained clearly and in detail.

One minor suggestion I have is really just a bugbear of mine. I prefer referring to the target antigen rather than the antibody name when describing immunoreactive staining. Therefore, I would prefer, in line 149, to refer to beta-tubulin expression rather than Tuj1 expression.

6. PLOS authors have the option to publish the peer review history of their article (what does this mean?). If published, this will include your full peer review and any attached files.

Reviewer #1: **Yes: **J Martin Collinson

Reviewer #2: No

---

## [Author Response · Author response to Decision Letter 0]

17 Oct 2022

Reviewer #1: This manuscript describes molecular and cellular analyses of mouse forebrain organoids, comparing wild-type with those derived from Pax6-null ES cells, and comparing wt and Pax6-null cells in conditional mosaics. In general the results are similar to those seen previously in vivo - that the patterning of the telencephalon is broadly OK in Pax6 mutants, but with evidence of a slower cell cycle in Pax6-null cells and premature exit from the cell cycle, disturbing the proliferation-differentiation balance. The authors argue that the similarity between mutation effect in vivo and in the organoids validates the use of organoids for developmental studies. The experiments have been performed and presented well and the differences between organoid batches treated and described carefully. I LOVE the segregation of mutant and wild-type cells into stripes in the mosaic organoids, which reflects the in vivo chimeras. I have some technical and other comments below.

We are grateful to the reviewer for his kind comments

Technical things to be addressed:

1) Nothing is said of the genetic background of the ES cells used to make the Pax6 organoids and whether they are the same as the control ES cells. If they are different, this could potentially undermine aspects of the quantitative analyses between the wt and null cell lines (in the same way that comparison between different mouse strains would be questionable). Authors should clarify and also describe genetic background of the conditional mutant cells.

Pax6-/- ES cells were on a 129Sv(Ola) background, Pax6cKO ES cells were on a mixed CD1/C57Bl6 background and BF1Venus cells were on EB3 genetic background, we have added this information to the Experimental procedures section (lines 431-7) and added a comment in the discussion section (lines 419-425).

We can’t rule out a possible contribution of genetic background to organoid phenotypes, but we consider that any such effects are likely to be minor, as Pax6-/- mutant mouse brain phenotypes are very consistent across multiple genetic backgrounds including CD1 (Estvill Torrus et al 2002) and C57BL/6J-DBA/2J (Kroll & O’Leary, 2005; Stoykova et al., 1996) and even the Pax6-/- rat phenotype is remarkably similar (Fukuda et al., 2000). We have cited these additional references. 

2) In Figure 2 and lines 171-172, the reduction of EdU-positive cells with Pax6 mutation has a fairly marginal level of significance (P=0.043), which the authors need to justify or comment on. Also in Fig 2C this difference is represented as **** which seems excessive - * at best. This could be in inter-strain difference if genetic backgrounds are different (see 1). In fig 2F/line 196 the P value of 0.023 is better, but could still be in the range of inter-strain differences raised in 1).

We thank the reviewer for pointing out our mistake, this was a transposition error when writing the original manuscript. In fact, the p value for the data shown in Fig. 2C was <0.0001, and is therefore highly significant. All original underlying data has been uploaded in Supplementary Table 2. 

3) Authors are to be commended for showing all data points in full, but the data in Fig 2F look weird to me. While there is a difference in mean % between wt and null, the data seem to consist of a lot of organoids where no abventricular mitoses occurred, and a lot where roughly similar (comparing wt and null) levels of mitoses happened, and one outlier at 100%. The significant difference could be due to different proportion of organoids scored (wt v Pax6) that had no abventricular mitoses. These data need more critical comment

We have added a more detailed description of the quantitation method in the Experimental procedures section of the manuscript, to help address this point (lines 498-503). We have also included the underlying data, in supplementary table 2, which clearly shows that there are many more abventricular mitoses in the Pax6-/- mutant organoids. As the reviewer correctly points out, there are a large number of organoid sections that contain no abventricular mitoses, especially in the wild type organoids. This leads to a large number of zeroes in the data, so the data is not normally distributed. We applied the appropriate statistical test for this situation, a non-parametric Mann-Whitney test. 

4) fig 3/4. It is difficult for the reader to compare colocalisation of GFP with nuclear markers e.g. EdU or with the Emx1 in situ. Can the authors present separate channels, and/or comment on how a magenta label was assigned to a patch of green GFP. The magenta channels in Fig 4B-F in particular are very weak, and I would have difficulty interpreting.

As requested by the reviewer we have added panels to Figure 3 to show the magenta and green channels separately, as well as together & updated the legend accordingly (lines 265-278).

We have removed the DAPI channel in Figure 4, to enhance the visibility of the magenta stain (EdU) to make the image easier to interpret.

5) Can authors present or describe (line 226) whether they observed leakiness of Cre activity in absence of tamoxifen.

We have added a line to indicate that no leakiness of cre activity was detected in the absence of tamoxifen addition (line 235-6).

6) No negative controls are shown or described for any immunos or the in situ - should be addressed.

We have added a sentence to the materials and methods to describe the negative controls used for immunostaining and in situs (lines 485-6).

Self-important reviewer comments that authors might want to comment on or address:

7) The argument that it is unclear whether organoid development reflects the developmental pathways observed in vivo (lines 62-63) seems weak to me. It is very well established that multiple organoid systems from mice and humans self-organise and develop in very similar ways using very similar developmental pathways to in vivo. Sure there are differences due to absence of vascularisation and reduced complexity of cell composition, but the validity of organoids to inform in vivo development is not really in question. This does not detract from the value of the current study.

We would prefer to leave this unchanged. There are unique challenges in studying human embryonic brain development directly, embryonic cell types can be characterised in great detail (for example by scRNAseq) but details of specific developmental mechanisms are much harder to unravel – hence the common use of model organisms. A unique advantage of mouse organoids is that we can compare the effects of specific mutations in vivo (in embryos) and in vitro (in organoids), to determine the extent to which molecular mechanisms are shared.

8) The authors comment on difference in cell cycle time and S-phase between organoids and in vivo. The discussion of regionalisation is important and valuable. Can the authors justify why they compare their data to E12.5 mouse and not later stages? Also I can understand why cell cycle time changes because cells might spend more time in G0/G1, but it is not clear to me why S-phase would increase - what is the mechanism for slowing down DNA replication? Is it firing of origins? It may not be relevant to current study, but authors could comment if they choose. Also the authors reference how Tc was calculated but the paper would benefit from description in methods.

We focussed our analysis on E12.5 as the Pax6-/- mutant mouse is well characterised at that time point. We agree that it would be interesting in future work to extend analysis of organoids to additional timepoints. We do not know what the mechanism underlying lengthening of S-phase is and agree that it is beyond the scope of the current study. We have previously found lengthening of S phase of Pax6-/- mutant cortical progenitors (Mi et al., 2018), so the finding is not unprecedented. 

Cumulative BrdU labelling is a long-established method of calculating cell cycle. We have cited the original paper that describes the technique and a more recent paper from our group which describes use of this method to calculate cell cycle lengths in Pax6 mutant mouse brains, readers interested to learn more about the method can consult these papers.

9) Line 336 the authors comment that it was unexpected for the 4-OH tamoxifen to produce a mosaic deletion, but I would be astonished if it didn't. As far as I can tell, Tmx always produces mosaicism and the people who claim 100% recombination are fooling themselves. The mosaicism is a strength of the current study and the authors would be justified in saying they planned for this all along.

We agree with this comment and have deleted ‘unexpectedly’

10) The morphology of the organoids is OK (adequate for the data), but I've seen a lot better. Do the authors want to comment on whether they consider their conditions optimised?

There are surprisingly few published studies involving mouse cerebral organoids to date, but we agree that their morphology is usually not as good as that of comparable human forebrain organoids. It is likely that the protocol could be improved in future.

Minor comments:

11) Red and magenta are poor choices for presenting two-colour data (Fig 2A,B).

We have changed magenta to white in these panels, to improve the clarity of the image.

12) There is a problem with labelling of Y-axis in Fig 4H

• This has been corrected

Reviewer #2: The authors state that data is fully available without restriction but they don't describe where or how the data can be accessed, as required in the data availability statement.

• We have now included supplementary tables containing the required data.

This manuscript describes some carefully conducted experiments with mouse organoids designed to demonstrate the suitability of these models for studying gene expression changes by comparing the effects with those seen in animals with such manipulations. The gene chosen is the well characterised transcription factor PAX6, and as this study comes from a Lab that has made many contributions to our understanding of the role of PAX6 in cortical development. They have been able to adapt experimental measures used in animal studies such as neuron production, and progenitor proliferation including rates of proliferation. I am satisfied that there conclusion that the behaviour of PAX6 expressing cells in both embryonic cortex in vivo and in organoid preparations share strong enough similarities to make use of organoids a useful model in this context. Some differences and unexpected results were also found, and these are explained clearly and in detail.

One minor suggestion I have is really just a bugbear of mine. I prefer referring to the target antigen rather than the antibody name when describing immunoreactive staining. Therefore, I would prefer, in line 149, to refer to beta-tubulin expression rather than Tuj1 expression.

• We have changed Tuj1 to �-tubulin throughout the manuscript.

---

## [Decision Letter · Decision Letter 1]

11 Nov 2022

Pax6 mutant cerebral organoids partially recapitulate phenotypes of Pax6 mutant mouse strains

PONE-D-22-20717R1

Dear Dr. Mason,

We’re pleased to inform you that your manuscript has been judged scientifically suitable for publication and will be formally accepted for publication once it meets all outstanding technical requirements.

Kind regards,

Michael Klymkowsky, Ph.D.

Academic Editor

PLOS ONE

Additional Editor Comments (optional):

Reviewers' comments:

Reviewer's Responses to Questions

**Comments to the Author**

1. If the authors have adequately addressed your comments raised in a previous round of review and you feel that this manuscript is now acceptable for publication, you may indicate that here to bypass the “Comments to the Author” section, enter your conflict of interest statement in the “Confidential to Editor” section, and submit your "Accept" recommendation.

Reviewer #1: All comments have been addressed

Reviewer #2: All comments have been addressed

2. Is the manuscript technically sound, and do the data support the conclusions?

Reviewer #1: Yes

Reviewer #2: Yes

3. Has the statistical analysis been performed appropriately and rigorously? 

Reviewer #1: Yes

Reviewer #2: Yes

4. Have the authors made all data underlying the findings in their manuscript fully available?

Reviewer #1: Yes

Reviewer #2: Yes

5. Is the manuscript presented in an intelligible fashion and written in standard English?

Reviewer #1: Yes

Reviewer #2: Yes

6. Review Comments to the Author

Reviewer #1: Thank you for addressing my comments. I am happy with the changes and look forward to seeing this in print.

Reviewer #2: (No Response)

7. PLOS authors have the option to publish the peer review history of their article (what does this mean?). If published, this will include your full peer review and any attached files.

Reviewer #1: No

Reviewer #2: No

---

## [Editor Report · Acceptance letter]

15 Nov 2022

PONE-D-22-20717R1 

*Pax6* mutant cerebral organoids partially recapitulate phenotypes of *Pax6* mutant mouse strains 

Dear Dr. Mason:

I'm pleased to inform you that your manuscript has been deemed suitable for publication in PLOS ONE. Congratulations! Your manuscript is now with our production department. 

Kind regards, 

on behalf of

Dr. Michael Klymkowsky 

Academic Editor

PLOS ONE